# AI-Predicted mTOR Inhibitor Reduces Cancer Cell Proliferation and Extends the Lifespan of *C. elegans*

**DOI:** 10.3390/ijms24097850

**Published:** 2023-04-25

**Authors:** Tinka Vidovic, Alexander Dakhovnik, Oleksii Hrabovskyi, Michael R. MacArthur, Collin Y. Ewald

**Affiliations:** 1Tinka Therapeutics, Fra Ivana Rozica 7, 21276 Vrgorac, Croatia; 2Laboratory of Extracellular Matrix Regeneration, Institute of Translational Medicine, Department of Health Sciences and Technology, ETH Zürich, CH-8603 Schwerzenbach, Switzerland; 3Palladin Institute of Biochemistry of the NAS of Ukraine, 02000 Kyiv, Ukraine; 4Lewis-Sigler Institute for Integrative Genomics, Princeton University, Princeton, NJ 08540, USA

**Keywords:** AI drug discovery, mTOR, rapalog, *C. elegans*, cancer, longevity

## Abstract

The mechanistic target of rapamycin (mTOR) kinase is one of the top drug targets for promoting health and lifespan extension. Besides rapamycin, only a few other mTOR inhibitors have been developed and shown to be capable of slowing aging. We used machine learning to predict novel small molecules targeting mTOR. We selected one small molecule, TKA001, based on in silico predictions of a high on-target probability, low toxicity, favorable physicochemical properties, and preferable ADMET profile. We modeled TKA001 binding in silico by molecular docking and molecular dynamics. TKA001 potently inhibits both TOR complex 1 and 2 signaling in vitro. Furthermore, TKA001 inhibits human cancer cell proliferation in vitro and extends the lifespan of *Caenorhabditis elegans*, suggesting that TKA001 is able to slow aging in vivo.

## 1. Introduction

One of the most robust interventions to increase healthspan and lifespan in preclinical models is the inhibition of the growth-regulating serine/threonine kinase mechanistic Target of Rapamycin (mTOR). This effect has been demonstrated using genetics in multiple species, including flies, worms, and mice. In *Drosophila,* the suppression of TOR signaling by overexpression of the negative regulators TSC1 or TSC2, or knock-in of dominant negative forms of TOR or S6K, significantly extended lifespan [1]. In *C. elegans,* rapamycin or RNAi-mediated knockdown in adulthood of the conserved TOR pathway components *daf-15* (RAPTOR), *rheb-1*, *raga-1*, or *ragc-1* all robustly extended lifespan [2,3,4,5,6]. In mice, while complete knockout of most mTOR components is embryonically lethal, knock-in of a hypomorphic mTOR allele extends the lifespan by ~25% [7].

The pharmacologic inhibition of mTOR has also been proven to be effective in extending the lifespan in mice. The mTOR inhibitor rapamycin has repeatedly shown lifespan-extending effects across sex, strain, and dosing regimens. Effective regimens include dosing continuously throughout life, intermittently throughout life, transiently early in life, transiently in midlife, and continuously starting late in life [8,9,10,11,12]. These striking preclinical effects are mediated both directly through the suppression of tumorigenesis and indirectly via the modulation of lifespan-regulating processes, including insulin and ATF4 signaling pathways [4,13,14].

Rapamycin and structurally related “rapalogs” have also shown promise in human trials against age-associated pathologies. In randomized clinical trials, the pharmacological inhibition of mTOR ameliorated the age-related decline of the immune system and reduced skin senescence in elderly humans [15,16].

Despite the clear preclinical promise of rapamycin and related rapalogs, challenges of drug discovery and development have been well-described, with cost estimates of over $100 million over ten years or more [17]. However, the use of artificial intelligence (AI) assisted methods for early drug discovery can fundamentally transform this process, cutting off years and tens of millions of dollars. We employed an AI method to identify a potent and selective mTOR inhibitor and validated this compound’s anti-cancer and pro-longevity effects in cell culture and *C. elegans* models.

## 2. Results

To identify and generate compounds that inhibit mTOR, we used generative adversarial networks and reinforcement learning methods. We generated more than 1000 small molecules predicted to target mTOR. We performed an independent validation using PASS software [18] to select which generated molecules have a high probability of inhibiting mTOR. We narrowed down the candidate list to 132 compounds with a high probability of targeting mTOR (Figure 1A). Since the PASS software can be used to predict toxic and adverse effects, we filtered these candidate compounds for their likelihood of low toxicity, of which 29 compounds remained (i.e., 22% of all 132 compounds; Figure 1A, Appendix A). Next, we assessed in silico these 29 compounds with preferable ADMET profiles (absorption, distribution, metabolism, excretion, and toxicity) and found one strong candidate 1-ethyl-3-(4-(4-morpholino-5,7-dihydrofuro[3,4-d]pyrimidin-2-yl)phenyl)urea, which we named TKA001 (Figure 1B–D, Appendix A).

To identify the binding pose of the TKA001 to mTOR kinase, we performed molecular docking using the X-ray structure (PDB ID: 4JT6) based on the fact that the kinase domain was co-crystalized with a compound in the ATP-binding site that contains a morpholine moiety presented in the ligand of interest (detailed approach is described in Methods and Materials). We performed docking with TKA001 and estimated a binding pose to the mTOR of ΔG = −8.44 kcal/mol (Figure 2A,B). We observed the formation of two hydrogen bonds between NH groups of urea moiety of the ligand and carboxyl group of Asp2195 as well as a hydrogen bond between the oxygen of the morpholine ring, which binds to the adenine pocket, and NH group of Val2240 (Figure 2C). Additional stabilization of the complex is provided by van der Waals and Pi–alkyl contacts between ligand atoms and amino acid residues (Figure 2C).

For further analysis of the protein–ligand complex stability as well as intermolecular interactions within it, we performed 100 ns molecular dynamics of the docked complex. We found that TKA001 binds tightly to an mTOR kinase based on a root mean square deviation (RMSD) value of 0.16 nm, which is below the 0.25 nm protein backbone RMSD value (Appendix A). Furthermore, using root mean square fluctuations (RMSF), we determined the local conformation changes in the protein chain during molecular dynamics. We identified twelve regions with RMSF values above 0.3 nm, including two regions with maximum RMSF values of 1.24 nm and 1.13 nm, which belong to two highly mobile unstructured parts of protein: Gly1822-Leu1865 and Asp2433-Asn2494 (Appendix A). TKA001 forms several hydrogen bonds with residues of the ATP-binding site, and we predicted the most common conformations of the protein–ligand complex (Figure 2D,E, more details in the Section 4).

Next, we assessed mTOR downstream signaling. The mTOR kinase is found in two complexes (mTORC1 and mTORC2 [19]). The mTOR complex 1 (mTORC1) phosphorylates ribosomal protein S6 kinase (S6K; [20]), whereas the mTOR complex 2 (mTORC2) phosphorylates AKT [20]. At 1 µM of TKA001, phosphorylation of S6K (S240/244) and AKT (S473) were reduced in HT1080 cells (Figure 3A), suggesting that TKA001 inhibits mTOR kinase in both complexes.

Our in silico analysis of TKA001 using CLC-Pred [21] predicted an 85% likelihood as an effective treatment against prostate cancer. We found that TKA001 inhibits cancer cell proliferation of epithelial cancer cells from patients with fibrosarcoma (HT-1080; half maximal inhibitory concentration (IC_50_) = 200 nM; Figure 3B) and cervical cancer cells (HeLa; IC_50_ = 1 µM; Figure 3C). In comparison, IC_50_ of rapamycin on HT-1080 or HeLa is 1.8 µM and 0.25 µM, respectively [22]. This suggests that TKA001 is a potent inhibitor of cancer cell proliferation in vitro.

Since TKA001 performed well in silico and in vitro, we next wanted to assess the in-vivo efficacy of TKA001 on mTOR inhibition. The genetic inhibition of mTORC1 or mTORC2, knockdown of mTOR/LET-363, or rapamycin treatment increases the lifespan of *C. elegans* [2,3,4,5,6,23,24,25] (Appendix A). To determine whether TKA001 could also increase the lifespan of *C. elegans*, we used 100 µM and 200 µM of TKA001 because about 100 µM of rapamycin results in the most robust lifespan extension [2,4,6]. We found that an adulthood-specific application of 200 µM of TKA001 only extended the maximal lifespan, whereas 100 µM of TKA001 resulted in both a mean and maximum increase of lifespan (Figure 3D, Appendix A). Next, we assessed whether a lower dose would be sufficient to increase lifespan. Indeed, supplementing *C. elegans* starting from the young adult stage with 10 µM of TKA001 was sufficient to extend the lifespan (Figure 3E, Appendix A).

## 3. Discussion

The nutrient-sensing mTOR kinase is a master growth regulator essential for development and tissue homeostasis [26]. However, mTOR activity becomes deregulated during aging, showing improper and sustained mTOR signaling in older animals [27,28]. Reducing the function of mTOR increases the lifespan in multiple organisms from yeast to mammals [29]. Despite mTOR being the prime target against many age-related and chronic pathologies [30], relatively few mTOR inhibitors have been developed to slow the aging process.

Using deep neuronal artificial learning, we identified many potential mTOR inhibitors (Appendix A). Through in silico analysis, we selected one mTOR inhibitor with predicted low toxicity and a preferable ADMET profile. Our modeling suggests on-target inhibition by molecular docking to the mTOR kinase. We confirmed these in silico observations by showing the inhibition of mTOR signaling and cancer cell proliferation in vitro, and increasing lifespan in vivo.

In a series of rational design and classical medicinal chemistry approaches, structurally similar mTOR inhibitors were previously developed based on a quaternary-substituted dihydrofuropyrimidine [31]. The structurally most similar compounds inhibited mTOR kinase signaling in vitro at 3–4.4 nM [31] and had an IC_50_ of 31–1700 nM in cancer cell proliferation assays (NCI-PC3, MCF7neo/Her2) [31] compared to our IC_50_ of 200 nM or 1000 nM in HT-1080 or HeLa, respectively. These comparable in vitro results validate our machine learning approach to identifying novel small molecules for mTOR inhibition. More importantly, we showed that our mTOR inhibitor TKA001 is able to slow aging in vivo. Further machine learning approaches hold the potential to speed up drug discovery and facilitate the selection of compounds for future clinical candidates targeting the mTOR-mediated healthy longevity benefits.

## 4. Materials and Methods

### 4.1. Reagents

TKA001 is 1-ethyl-3-(4-(4-morpholino-5,7-dihydrofuro[3,4-d]pyrimidin-2-yl)phenyl)urea (C_19_H_23_N_5_O_3_). Molecular weight: 369.43, Melting point 213 °C. TKA001 was synthesized by Otava Chemicals (Concord, ON, Canada). ZINC71297044 (catalog number 27705871).

Dimethyl sulphoxide (DMSO) CAS# 67-68-5 BDH Chemicals (VWR, Dietikon, Zurich, Switzerland) 500 mL, analytical reagent. Rapamycin (LC Laboratories, Woburn, MA, USA, Cat. No. R-5000).

### 4.2. Machine Learning

The list of molecules that are active and inactive on the mTOR target was downloaded from the ExCAPE database [32]. There were in total 46,679 molecules, and 2878 molecules were classified as active on the mTOR kinase. We divided these molecules to a test and training set, and the molecules were converted to an ECFP6 fingerprint with 2048 bits using RDkit v2023 (https://www.rdkit.org, accessed on 21 January 2023). Random forest (RF) classifier from scikit-learn [33] was used to train and discriminate between active and inactive mTOR molecules. The generative adversarial networks and reinforcement learning methods were used to generate novel molecules that potentially can modulate the activity of mTOR. Predicted activity on the mTOR target and a quantitative estimate of the drug-likeness (QED) score were used to evaluate generated compounds.

### 4.3. In Silico Prediction of the Mechanism of Action and Toxicity of Candidate Compounds

The online web tool PASS v2.0 [18] was used to predict the pharmacological activities of compounds, as well as their toxicity. PASS indicates the probable activity (Pa) and probable inactivity (Pi) of ‘drug-like’ substances. Using PASS v2.0 (https://www.way2drug.com/PASSOnline/index.php, accessed on 21 March 2023), it is possible to obtain an estimated biological activity profile of a drug-like molecule using only structural formulas. Some of the predicted activities of PASS software version 2.0 are pharmacological effects, mechanism of action, as well as toxic and adverse effects.

### 4.4. In Silico Predicted Physicochemical Properties of TKA001

ADMETlab v2.0 [34] was used to predict the physicochemical properties, drug-likeness, and toxicity of TKA001.

### 4.5. Molecular Docking

The structure of TKA001 was drawn using the MarvinSketch v21.16.0 software, after which the ligand was protonated, and its low-energy conformations were generated (MarvinSketch version 21.16.0, ChemAxon (https://www.chemaxon.com, accessed on 12 April 2023)). The crystal structure of mTOR kinase (PDB ID:4JT6) was downloaded from RCSB Protein Data Bank [35]. Using UCSF Chimera 1.16, the protein structure was prepared for molecular docking by removing all chains except chain B and adding hydrogens with subsequent energy minimization of the protein–ligand complex [36]. After that, the co-crystalized ligand was removed. A molecular docking simulation of protein and ligand was performed using SwissDock web server, which uses the protein–ligand docking program EADock DSS v3 [37]. A search space of 20 × 20 × 20 Å was used with a grid box centered on the ATP-binding pocket (X center: −18.24, Y center: −33.14, Z center: −55.23). Additionally, an accurate docking type was selected with the following parameters: 5000 binding modes were generated by the DSS engine, relaxation of the most favorable binding modes was performed by 100 minimization steps using the steepest descent algorithm followed by 250 steps of adopted-basis Newton–Raphson (ABNR) minimization, and 250 binding modes were clustered and evaluated by the FullFitness [38]. Interaction types were evaluated, and images were made with Discovery Studio Visualizer (BIOVIA, Dassault Systèmes, Discovery Studio Visualizer, v.16.1.0, San Diego, CA, USA: Dassault Systèmes, 2015). The final choice of the most promising binding pose was based on the estimated ΔG in kcal/mol, the size of the clusters, and the number of hydrogen bonds formed between the protein and ligand.

### 4.6. Molecular Dynamics

A molecular dynamics (MD) simulation was used to estimate ligand stability in the binding pocket as well as to assess crucial interactions of the protein–ligand complex obtained by molecular docking. The calculations were carried out using Gromacs 2020.6 software [39] in Charmm36 force field [40].

Before proceeding to molecular dynamics, the missing regions (K1815-Q1866 and K2437-E2491) of mTOR kinase (PDB ID: 4JT6) were built using the Modeller software v10.4 [41]. In the newly built model, the coordinates of amino acid residues remained the same as in the crystallographic structure, while missing loops were built using the AlphaFold2-generated model as templates.

After that, the mTOR kinase was protonated according to the build-in function in Gromacs v4.5.3. The topology file for the ligand was generated by SwissParam [42]. The complex was placed into the center of a periodic dodecahedron box which was filled with TIP3P water molecules. The minimum distance between the edge of the simulation box and the complex was 1.0 nm. Na^+^ and Cl^−^ ions were added to neutralize the system electrostatically with the final 120 mM ionic concentration. To relieve any possible steric clashes and inappropriate geometry, energy minimization of the complex was performed. The system was relaxed by applying the steepest descent algorithm (the maximum number of steps was 50,000). Then, the equilibration was conducted in two phases: under an NVT ensemble for 100 ps and under an NPT ensemble for 1 ns using the modified Berendsen (V-rescale) thermostat and Parrinello–Rahman barostat. After that, we launched the MD simulation. All calculations were performed using the “leap-frog integrator” at the temperature of 300 K and at a constant pressure (1 Bar).

The trajectories were analyzed using Gromacs analysis tools. The obtained results were plotted by using the XMGrace v5.1.25 software. The percentage of the existence of intermolecular hydrogen bonds was calculated with the Perl script plot_hbmap.pl provided by Prof. Justin Lemkul.

### 4.7. Detailed Analysis of Molecular Docking and Molecular Dynamics

To identify the binding pose of the TKA001-protein system, we performed molecular docking using the SwissDock server. We chose the X-ray structure (PDB ID: 4JT6) based on the fact that the kinase was co-crystalized with a compound in the ATP-binding site that contains a morpholine moiety presented in the ligand of interest. Moreover, the presence of the ligand optimizes the position of the amino acid residue side chains, which may lead to more precise results.

To evaluate the accuracy and ability to reproduce co-crystalized binding geometry and orientation of ligands by the SwissDock engine for particular proteins and ligands, we performed re-docking. The co-crystalized ligand was retrieved and placed in a random position and orientation in the space. After that, molecular docking was conducted. According to the results, the binding pose, which belonged to the first cluster with a population of eight members and estimated ΔG = −9.33 kcal/mol after superimposing by SMARTS with crystal conformer, had RMSD values of 0.8095 Å, which are considered appropriate.

The results of the molecular docking of the mTOR kinase and TKA001 using SwissDock are shown in Figure 2A,B. The binding pose of the ligand, which had an estimated binding affinity ΔG = −8.44 kcal/mol, was used for analysis. We observed the formation of two hydrogen bonds between NH groups of urea moiety of the ligand and carboxyl group of Asp2195 as well as a hydrogen bond between the oxygen of the morpholine ring, which binds to adenine pocket, and NH group of Val2240. Additional stabilization of the complex is provided by van der Waals and Pi–alkyl contacts between ligand atoms and amino acid residues presented in Figure 2C.

For further analysis of the protein–ligand complex stability as well as intermolecular interactions within it, we performed 100 ns molecular dynamics of the docked complex. The dynamic stability of the protein–ligand complex was assessed by using calculated root mean square deviation (RMSD) changes during a full molecular dynamics simulation except for the first 5 ns. For both the protein and ligand RMSD calculations, the range of residues from 2000 to 2400, which includes residues of catalytic cleft, was used for the least squares fit as the presence of two long and highly mobile unstructured regions skew the results. Appendix A displayed some minor fluctuation in protein RMSD that remained stable for 100 ns and had an average value of 0.2 nm (black line), while the average RMSD value for TKA001 is below 0.16 nm (red line). Minor ligand fluctuations may be caused by forming–breaking H-bonds during simulation. However, since the interaction between the ligand and protein is considered stable when the RMSD value is under 0.25 nm, we assume that the TKA001 binds tightly to mTOR kinase.

Root mean square fluctuations (RMSF) were studied to characterize and determine the local conformation changes in the protein chain during molecular dynamics. As shown in Appendix A, there were twelve regions with RMSF values above 0.3 nm, including two regions with maximum RMSF values of 1.24 nm and 1.13 nm, which belong to two highly mobile unstructured parts of protein: Gly1822-Leu1865 and Asp2433-Asn2494. However, amino acid residues that belong to the ATP-binding site have RMSF values below 0.25 nm which implies stability and rigidity of the ligand binding site in the presence of the latter.

According to the results of molecular dynamics, the TKA001 forms several hydrogen bonds with residues of the ATP-binding site. Around 74% of the simulation time, there exists a hydrogen bond between the oxygen of the morpholine ring and NH group of Val2240, around 48% and 55%, and around 74% and 67%—between the first and the second NH groups of urea moiety and C=O, C–O groups of Asp2195, respectively. Surprisingly, we observed the presence of another weak hydrogen bond between C=O group of urea moiety and NH group of Phe2358 for around 51% of the time, which was not determined earlier by molecular docking. In addition to the presence of numerous weak Pi–alkyl contacts, we detected the existence of Pi–Cation interactions between the benzene ring of the ligand and Lys2187, as well as the non-durable formation of a hydrogen bond between the NH3 group of Lys2187 and C=O group of ligand (time of existence was less than 1 ns). The most common conformations of the protein–ligand complex are shown in Figure 2D,E.

### 4.8. Human Cell Lines

Cell lines were maintained at 37 °C in a 5% CO_2_ incubator. Cells were maintained in Dulbecco’s Modified Eagle Medium (DMEM, Thermo Fisher Scientific, Waltham, MA, USA) supplemented with 10% Hyclone Cosmic Calf Serum (Cytiva, Marlborough, MA, USA) and 1% penicillin-streptomycin (Thermo Fisher Scientific). Cells were routinely passaged at 70% confluence. The following cell lines of the American Type Culture Collection (ATCC) were used: HT1080: ATCC CCL-121, COSMIC Database, sample ID 907064. https://cancer.sanger.ac.uk/cell_lines/sample/overview?id=907064, accessed on 14 February 2023 and HeLa: ATCC CRM-CCL-2, COSMIC Database, sample ID 1298134. https://cancer.sanger.ac.uk/cell_lines/sample/overview?id=1298134, accessed on 14 February 2023.

### 4.9. Human Cell Proliferation

For the proliferation assays, cells were seeded in 96-well plates at 2000 cells per well and treated with a vehicle (0.1% DMSO) or TKA001. After 24 h, cells were stained with Hoechst 33342 (Thermo Fisher Scientific) at 10 µg/mL, followed by imaging using a Cytation 5 Cell Imaging Multimode Reader (BioTek, Winooski, VT, USA). Cell count was performed by automated nuclei counting using Gen5 Data Analysis Software v2.0 (BioTek).

### 4.10. Western Blots

For Western blots, cells were grown to 70% confluence in 6-well plates. Cells were then treated with a vehicle (0.1% DMSO) or TKA001 for 2.5 h. Cells were then placed on ice, washed with cold phosphate-buffered saline (PBS), then harvested in 60 µL of cold RIPA buffer containing protease/phosphatase inhibitors (Thermo Fisher Scientific). Cells were incubated on ice for 20 min, then centrifuged at 20,000× *g* for 10 min, and supernatants were collected. Supernatants were standardized to a concentration of 2 mg/mL and combined with 6× loading buffer containing SDS, glycerol, beta-mercaptoethanol, and bromophenol blue. Samples were boiled at 100 °C for 10 min, and 15 µL was loaded on a Tris/glycine SDS-polyacrylamide gel. Proteins were separated by SDS-PAGE, transferred to a polyvinylidene difluoride (PVDF) membrane, and immunoblotted with indicated antibodies. The primary antibodies used were pS6 (S240/244): CST 5364, p-AKT (S473): CST 4060, ATK: CST 4691, and Vinculin: CST 13901. The secondary antibody was HRP-linked anti-rabbit IgG (CST 7074). Immunoblots were imaged using an enhanced chemiluminescent detection kit (ECL, Bio-Rad, Hercules, CA, USA) and visualized on a LiCor Odyssey Fx Imaging System (LiCor, Lincoln, NE, USA).

### 4.11. C. elegans Lifespan

TJ1060 [*spe-9(hc88)*; *rrf-3(b26)*] *C. elegans* were grown until the gravid adult stage and synchronized by bleaching [43]. Hatched overnight cultures of L1 *C. elegans* were seeded onto plates covered with heat-inactivated OP50 bacteria and left at a +25 °C incubator for 2 days to induce temperature-sensitive *spe-9(hc88)* mutation to achieve sterility of eggs. Then, young adult animals were transferred manually by picking 70–75 individuals onto two 6 cm plates (*n* ≥ 140 animals per condition) with corresponding TKA001 concentrations in both agar and heat-inactivated OP50. A further lifespan assay was performed at 20 °C until death [44]. The TKA001 was dissolved in DMSO, resulting in a 0.2% DMSO final concentration in the NGM plates, and, thus, 0.2% DMSO was used as the empty vehicle control. All the plates were normalized by 0.2% DMSO to exclude differences in solvent concentration. Death events were counted every second day and starting from adulthood day 12 (AD12) on a daily basis. *C. elegans* with a vulval protrusion, matricide events, dried on the walls, and abnormally looking were censored [45]. A Kaplan–Meier estimator was used for analysis in the GraphPad Prism 8 software. A log rank (Mantel–Cox) test was utilized for statistical analysis (*n* ≥ 100). Figure 3D: control = 128 death events, TKA001 (100 µM) = 132 death events [+8.9% mean lifespan increase], TKA001 (200 µM) = 132 death events [+4.7% mean lifespan increase]. Figure 3E: control = 137 death events, TKA001 (10 µM) = 131 death events [+6.3% mean lifespan increase]. For more details and statistics, see Appendix A.

### 4.12. C. elegans Lifespan Assessed with the Lifespan Machine

The setup and lifespan assays were performed as described in Statzer et al., 2022 [46]. In brief, wild type (N2) *C. elegans* were synchronized via bleaching and seeded onto 10 cm OP50-seeded plates and left for 48 h at 20 °C. After reaching the L4 developmental stage, *C. elegans* were shifted to plates containing 100 µM FUdR, ampicillin, and nystatin, seeded with heat-inactivated OP50 bacteria, with corresponding compounds priorly diluted in DMSO (0.2% concentration in the final NGM solution, including control). At adulthood day 12, animals were manually picked to lifespan plates with sealer, supplemented with the same compounds and heat-inactivated OP50 bacteria, and loaded into the lifespan machine [47]. Experiments were conducted in a controlled environment inside ventilated incubators at 18 °C. Raw data from the machine were plotted with JMP 16 software.

## Figures and Tables

**Figure 1 ijms-24-07850-f001:**
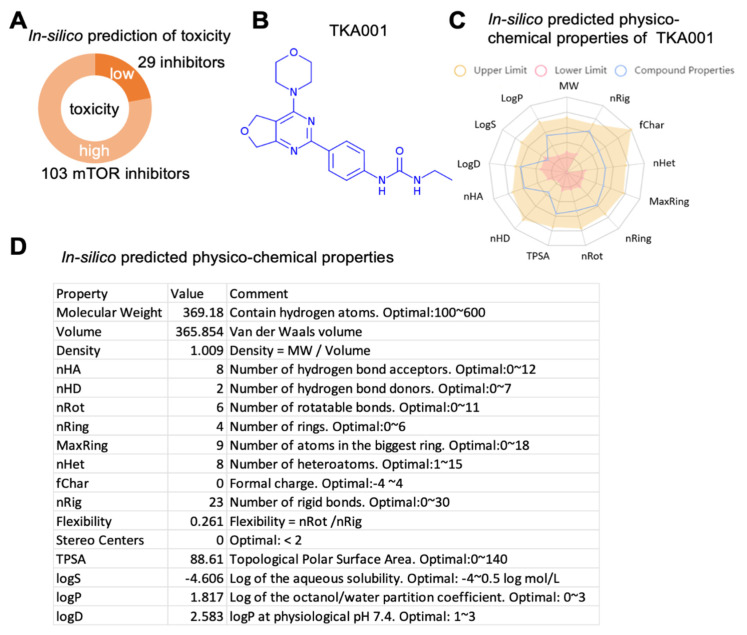
Characterization of mTOR inhibitor TKA001. (**A**) In silico prediction of toxicity of mTOR inhibitor candidate compounds (all predicted compounds are in Appendix A). (**B**) Lead candidate TKA001 (C_19_H_23_N_5_O_3_). (**C**,**D**) In silico-predicted physiochemical properties of TKA001. More details are shown in Appendix A.

**Figure 2 ijms-24-07850-f002:**
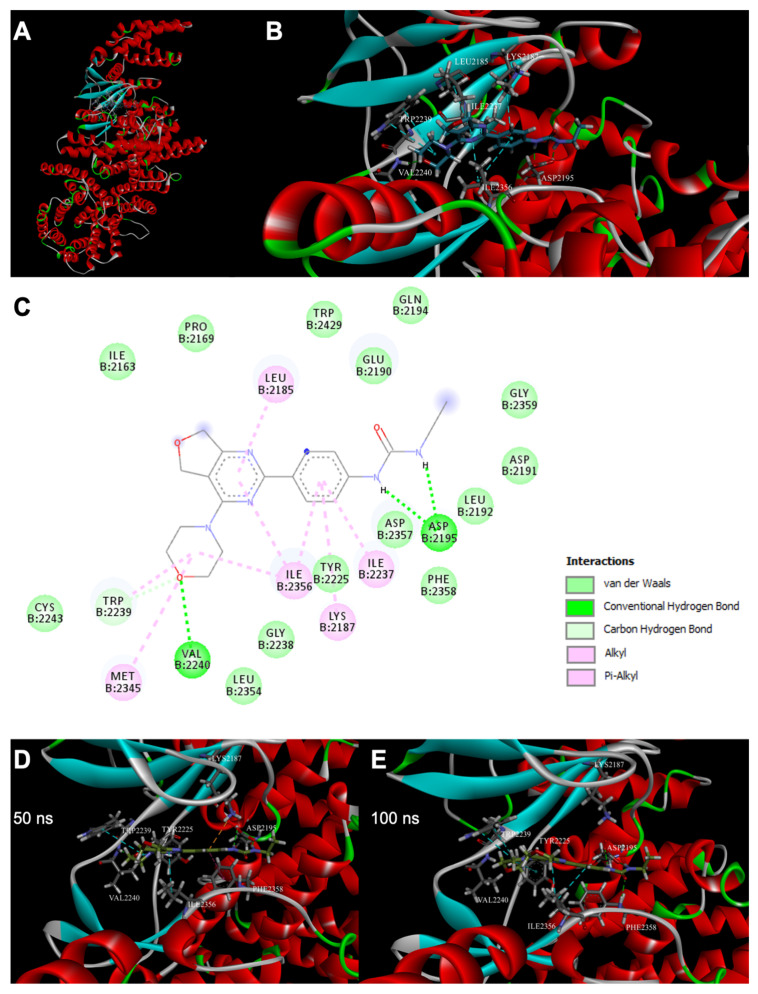
Molecular docking of mTOR with TKA001. (**A**) Global view of mTOR kinase (PDB ID:4JT6) with docked TKA001 ligand in the ATP-binding site. (**B**) Detailed view of mTOR-TKA001 complex centered on the mTOR catalytic cleft obtained by molecular docking. Stick representation of TKA001 (C, dark grey; O, red; N, blue; and H, light grey) and of mTOR residues that took part in interactions. Green dash lines indicated hydrogen bonds, cyan dash lines—Pi–Alkyl contacts. (**C**) mTOR-TKA001 interactions on a 2D diagram. Several types of interaction were shown by color with different amino acid residues. (**D**,**E**) The snapshots of the ligand–protein complex from the 100 ns molecular dynamics simulation. Shown at 50 ns (**D**) and at 100 ns (**E**). Stick representation of TKA001 (C, dark grey; O, red; N, blue; H, light grey) and of mTOR residues that took part in interactions. Green dash lines indicated hydrogen bonds, cyan dash lines—Pi–Alkyl contacts, orange dash lines—Pi–Cation contacts, magenta dash lines—Pi–Pi contacts.

**Figure 3 ijms-24-07850-f003:**
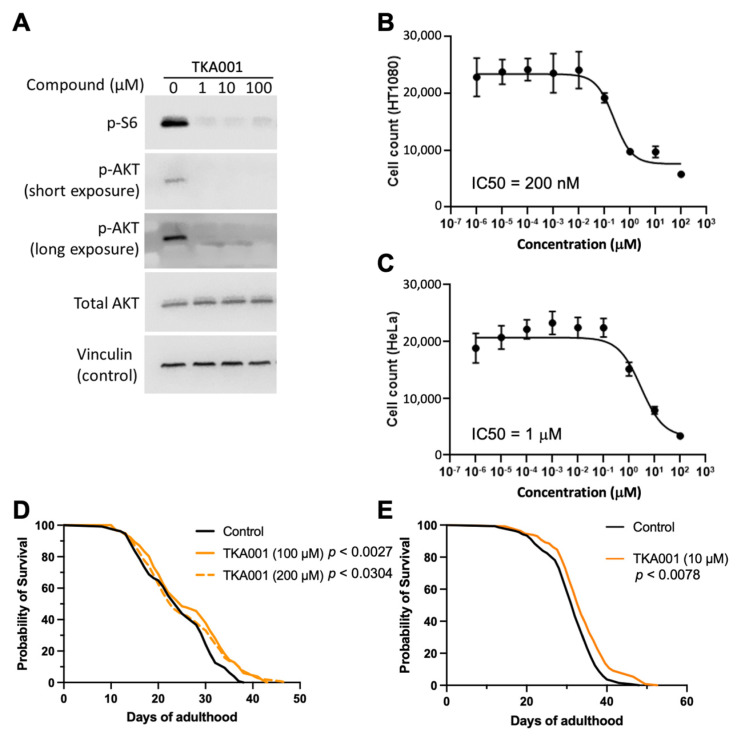
In vitro and in vivo validation of mTOR inhibitor TKA001. (**A**) Western blot of HT1080 cells pre-treated with increasing concentrations of TKA001. p-S6 indicates phospho-S6 (S240/244), and p-AKT indicates phospho-AKT (S473). The blots are shown as short and long exposure times, total AKT, and vinculin loading control. (**B**) Proliferation assays of TKA001 treatment on HT-1080 cells. Half maximal inhibitory concentration (IC_50_) = 200 nM. (**C**) Proliferation assays of TKA001 treatment on HeLa cells. IC_50_ = 1 µM. (**D**,**E**) Feeding 10, 100, or 200 µM TKA001 increased the lifespan of *C. elegans*. Two independent biological trials. Control = 0.2% DMSO. *p*-value determined with Log Rank. Raw data, statistics, and additional trials are in Appendix A.

## Data Availability

All data is available in the Appendix A.

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
