# Peer review of "AI-Predicted mTOR Inhibitor Reduces Cancer Cell Proliferation and Extends the Lifespan of C. elegans"

_ijms, 2023, doi:10.3390/ijms24097850_

Round 1

Reviewer 1 Report

The manuscript superficially describes selection and characterization of a compound that inhibits mTOR. An important aspect of the paper is that the compound was selected using "machine learning." However, there appears to be no description of this ML. The authors might just as well have said that they randomly chose ZINC71297044 from the zinc collection as their starting point.

Given that the text states that structure-based methods were used to "validate" the compound, it is surprising that there is nearly no description of how the compound is predicted to bind to mTOR. The authors said that they used the structure of 4JVS as the receptor in their docking studies for the ligand. However, the authors of 4JSV describe this a a hyperactive mTOR construct. It is surprising that the authors did not use the WT structure, 6BCX. While they say that they used PyRx for the docking, there is no explanation of how the docking was done. Was it flexible ligand? Was it flexible receptor? What was the pocket definition? In the methods, they state that "Adequate spacing between the grids was ensured so that the ligands could move freely inside.” Inside what? Inside the ATP-binding pocket? Was there really space for the compound to rotate end-for-end inside the pocket? It does not seem as though this would be possible. Was the morpholine moiety interacting with the hinge as is typical for the PI3Ks, or was it directed out of the pocket, as would be more typical for protein kinases? Was entire 4JSV used in the docking? What was the distance cutoff used for interactions? 

The authors state that after docking, they used PyMol for further analysis, yet there appears to be no analysis of the docked structure reported. PyMol is primarily intended for graphic representation of structures, with only rudimentary analysis capability. The only structure figure is 1E, which has so little detail that it is nearly useless. It seems as though the authors have removed parts of the structure, introduced heavy fog and clipping to obscure understanding even the orientation of the mTOR.  This needs to be labelled with residue numbers, showing side-chains in the pocket and labelling secondary structure elements (using the names in the paper describing 4JSV). In 1D there is a list of properties of the molecule that the legend refers to as "predicted" properties. Among these "predicted" properties are many that directly and trivially follow from the structure of the ligand as depicted in 1B, such as "number of rings",  such as "number of atoms in the biggest ring", etc.  

In Fig. 2A, there appears to be no difference between control and UV-B treated cells in the absence of the compound. This does not agree with other reports that show UV-B induced activation of mTOR. Indeed the authors state: "“To stimulate mTOR signaling, we UV-B irradiated HT1080 cells..." Confusingly, they state that they  "still found a strong reduction of S6K and AKT phosphorylation, suggesting that TKA001 efficiently reduces mTOR signaling (Figure 2A).” The figure does seem to show that the compound reduces mTORC1 and mTORC2 signalling, but it also shows that UV-B irradiation is inconsequential. Even more confusingly, there are the labels "light" and "dark" on whole lanes of the gel. What is "light" in the control for UV-B irradiation? What is "dark" in the UV-B irradiated lanes? There are no hints in the methods.

They state that they "confirm" their prediction using ”PASS”, described in 10.1093/bioinformatics/16.8.747. However, the online interface to which reference [18] refers simply gives “page not found” (in Russian).

The worm survival curves shown in 2D and 2E show much longer maximal survival even for the control than was observed in the worm study that the authors cite for rapamycin-induced longevity. In the study cited, rapamycin had a much greater influence than TKA001. There does not seem to be a clear pattern of TKA001 dose response on survival. 10 uM produces maximal longevity, followed by 200 uM then 100 uM. 

There is the statement "Worms with a vulval protrusion, matricide events, dried on the walls, and abnormally looking were censored ." It is indeed difficult to score worms reliably because of these events. The danger is that some subjectivity can be introduced. For this reason, it is important that the authors state whether observations were blinded with regard to the treatment. 

In the abstract, there is the statement that "We confirmed TKA001 binding in silico by molecular docking." Sadly, in silico docking never confirms anything. Binding measurements confirm binding. This statement is jarringly naive.

Reviewer 2 Report

Review of T. Vidovic et al., AI-predicted mTOR inhibitor reduces cancer cell proliferation and extends the lifespan of C. elegans, submitted to International Journal of Molecular Sciences

General Comments

This interesting paper by T. Vidovic et al. describes work using a series of in silico techniques to identify a likely candidate TOR inhibitor which they call TKA001.

They next show that TKA001 leads to lowered phosphorylation of both S6K and AKT, consistent with the possibility that it is an inhibitor of mTORC1 and mTORC2.

The authors then demonstrate that TKA001 inhibits cancer cell proliferation of epithelial cancer cells from patients with a fibrosarcoma.

Lowered TOR activity has been shown to increase overall lifespan in several model organisms, including the nematode Caenorhabditis elegans. The authors show that TKA001 can lead to a very modest but statistically significant increase in lifespan in C. elegans.

In Figure 1 D and Figure 1 E the mean lifespan of control and all treated worms should be shown on the graph, and the number of uncensored worms scored per treatment should be shown in the figure or described directly below in the caption. While the authors provide all of this information in the materials and methods section, in these panels a reasonably modest effect size is being described, further emphasizing the need for this information to be easily accessible to the reader in the figure itself.

Overall, this study describes findings of great interest to the field. The manuscript is very clearly written overall, with only one minor suggested correction, as noted below. The figures are also very clear and all add greatly to the manuscript.

Specific Comments

1. References. Appropriate italicization (e.g. of species, gene names, etc.) is missing from within the article titles of the references. This is often an artifact introduced by reference management software but should be manually corrected in the final version.

p { color: #000000; orphans: 0; widows: 0; margin-bottom: 0.08in; direction: ltr; background: transparent }p.western { font-size: 12pt; so-language: en-US }p.cjk { font-family: "WenQuanYi Zen Hei"; font-size: 12pt; so-language: zh-CN }p.ctl { font-family: "FreeSans"; font-size: 12pt; so-language: hi-IN }a:link { color: #0000ff; so-language: zxx; text-decoration: underline }

Round 2

Reviewer 1 Report

The revised manuscript has clarified the ML approach that the authors used to choose putative mTOR inhibitors. It also has a greatly improved structural description of the inhibitor/enzyme binding. These details are not just cosmetic. They will help set a standard for future approaches. The results are thoroughly described, and the outcome of the study is an interesting one. The following includes a clarification and a suggestion. I leave it to the discretion of the authors as to whether they follow the suggestion.

In the authors responses, they indicate that they have chosen a different starting model for their modelling of the enzyme/inhibitor complex, and that they now chose the structure of the complex of mTOR with PI-103, the pan-PI3K/mTOR inhibitor. This is a reasonable starting point. However, the authors seem to have mis-understood my original concern about 4JVS as a starting model. They responded saying “…The sequence of mTOR in the 4JT6 structure, as well as, in 4JVS is identical to the sequence of wild-type kinase and 6BCX. It can be simply checked by using sequence alignment between 4JT6 and the sequence from UniProtKB entry: P42345.” I know that the sequence for what there is of 4JVS is the same as the sequence of full-length mTOR as found in 6BCX. However, the truncation construct and the lack of a RAPTOR subunit for the crystal structures. entries 4JVS and 4JTS, results in a conformational change in the active site relative to the full-length mTORC1 (entry 6BCX), and the crystal construct has a much greater enzymatic activity than WT mTORC1. I was questioning whether they might have started with the WT enzyme conformation of mTOR by choosing 6BCX, instead of a hyperactive crystal construct, given that both structures were available, and the ultimate objective was to inhibit mTOR in cells with full-length WT mTORC1. It is a small point, and the current manuscript has gone to great lengths to characterise a likely mTOR/inhibitor structure. I only wanted to clarify the point I had originally raised.

I still find figure 2A confusing. There is no stimulation of mTORC1 activity by UV-B in the control. In their response to the review, the authors acknowledge that UV-B did not stimulate the control. Nevertheless, the text has not changed over what it was previously. They still say they are looking at the effect of the inhibitor in UV-B activated cells. A reader might see the results in 2A and think that UV-B is modifying some of the inhibitor to convert it to an activator so that the enzyme activity is increasing with increasing inhibitor concentration. I think this is unlikely but given the lack of UV-B effect on the controls, this is the kind of explanation that might be necessary. Are there multiple biological replicates for this experiment? There is no statement of this in the text. If not, maybe the best approach would be to remove the UV-B results.
